# Uncertainty Guided Global Memory Improves Multi-Hop Question Answering

**Alsu Sagirova**
Moscow Institute of
Physics and Technology,
Dolgoprudny, Russia
alsu.sagirova@phystech.edu

**Mikhail Burtsev**
London Institute for
Mathematical Sciences,
London, UK
mb@lims.ac.uk

## Abstract

Transformers have become the gold standard for many natural language processing tasks and, in particular, for multi-hop question answering (MHQA). This task includes processing a long document and reasoning over the multiple parts of it. The landscape of MHQA approaches can be classified into two primary categories. The first group focuses on extracting supporting evidence, thereby constraining the QA model's context to predicted facts. Conversely, the second group relies on the attention mechanism of the long input encoding model to facilitate multi-hop reasoning. However, attention-based token representations lack explicit global contextual information to connect reasoning steps. To address these issues, we propose GEMFormer, a two-stage method that first collects relevant information over the entire document to the memory and then combines it with local context to solve the task[1]. Our experimental results show that fine-tuning a pre-trained model with memory-augmented input, including the most certain global elements, improves the model's performance on three MHQA datasets compared to the baseline. We also found that the global explicit memory contains information from supporting facts required for the correct answer.

## 1 Introduction

Transformer (Vaswani et al., 2017) and its variants (Lin et al., 2022) have become one of the most popular solutions for various NLP tasks. Particularly, Transformers are applied to solve the multi-hop question answering (MHQA) tasks (Tu et al., 2019; Zemlyanskiy et al., 2021; Khattab et al., 2021) that require reasoning over multiple parts of the long document to answer the question.

The problem of reasoning in MHQA has attracted a lot of research recently (Mavi et al., 2022). One group of methods focuses on the usage of sub-networks or dedicated modules to extract evidence from a long document and then solve the question-answering (QA) task based on the detected evidence facts (Nishida et al., 2021, 2019; Tu et al., 2019; Bhargav et al., 2020; Zhao et al., 2023). The resulting performance of such models highly depends on the evidence extraction method quality, limiting the QA model context to a small number of pre-selected facts. Another group of methods addresses the general long document encoding problem by sparsifying the attention patterns to enlarge the maximal input sequence length (Beltagy et al., 2020; Ainslie et al., 2020; Zaheer et al., 2020). Despite the merits of these models, their attention-based token representations combine local and global information in the same vector. The high-level contextual features are spread over a long sequence which makes it harder to access them. To address the described problems, we propose GEMFormer (Global Explicit Memory Transformer). GEMFormer is a method for augmenting the pre-trained language model with memory to store global information relevant to the task. It processes long input concatenated with a memory sequence consisting of tokens from input that are important to solve the task. Token importance is defined by the language model uncertainty.

## 2 Related work

Augmentation of the neural network model with memory provides additional space to store relevant information that can be used to improve model performance and reduce computational costs. Early examples of memory-augmented neural network architectures such as RNN and LSTM (Hochreiter and Schmidhuber, 1997) used hidden states as internal memory. Graves et al. (2014) and Graves et al. (2016) introduced the external type of memory, where a separate network manipulated memory. With the growing popularity of the attention mechanism, attention was adopted for model-memory interaction (Weston et al., 2015; Sukhbaatar et al.,

---

[1] https://github.com/Aloriosa/GEMFormer

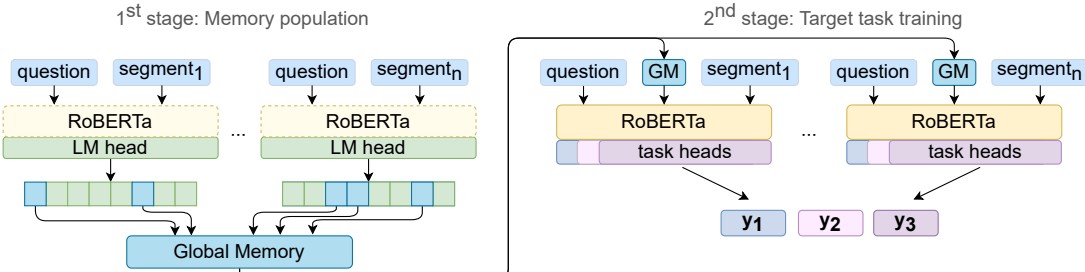

Figure 1: **Global Explicit Memory Transformer.** The input segments are processed by RoBERTa+LM head to generate prediction distributions for uncertainty estimation. Global memory is populated based on the given entropy condition. Then, the memory-augmented segments question+memory+context are used to obtain the MHQA predictions $y_1, y_2, y_3$.

2015; Chandar et al., 2016).

A number of recent papers used memory to store global input representations. Gupta and Berant (2020); Zemlyanskiy et al. (2021); Wu et al. (2022) employed memory to encode and retrieve the compressed view of the long input. Memory (Burtsev and Sapunov, 2020) and Recurrent Memory (Bulatov et al., 2022) Transformers used memory to store non-local representations of the processed sequence. The interpretable working memory in the Transformer decoder (Sagirova and Burtsev, 2022a,b) was presented to store contextual information that was not explicitly stated in the input. Sorokin et al. (2022) summarized representations of events from the distant past into memory to improve the model predictions. GEMFormer adapts the idea of interpretable memory by storing tokens from the input sequence. This augmentation of memory enriches long inputs with essential global contextual information important for the task.

## 3 Global Explicit Memory

We implemented GEMFormer with a RoBERTa-base (Liu et al., 2019) model as a backbone. Global explicit memory is a sequence of document tokens that are important for correct reasoning and answer prediction. The model's uncertainty is used to measure the importance of inputs. Given an input sequence $x = [t_1, t_2, \ldots, t_m]$ of $m$ tokens and a vocabulary of size $n$, we first obtain a vector $p = Softmax(LM\_RoBERTa(x))$ of token candidates' probabilities using RoBERTa with language modeling (LM) head. Then we calculate the entropy $H = -\frac{1}{n}\sum_{j=1}^{n} p_j \log p_j$ for each input position. In our experiments, we tested two

memory population conditions for entropy:

$$
\begin{aligned}
Highest\ H &= \arg \text{top } k\ [H(t_i), t_i \in x], \\
Low\ H &= [t_i \in x : H(t_i) < \theta],
\end{aligned} \quad (1)
$$

where $\theta$ is a model uncertainty threshold and $k$ is a selected memory size (see details in Section 4). The general motivation behind the entropy-based memory population is the following. The model is fed with a question and a context, and the entropy is computed for each contextual token. This entropy is conditional with respect to the question and token-surrounding context. Such entropy value measures how much entropy a token has remaining if we have already learned the question and the context. In other words, how easily the token can be predicted given the question and the context. Taking into account that a document is a collection of question-relevant and distractor paragraphs, the entropy of a task-relevant token should be lower than the entropy of the irrelevant one.

The GEMFormer architecture is depicted in Figure 1. To fit RoBERTa maximum sequence length limit, the contextual document is split into segments and each segment is concatenated to the question. Input processing consists of two stages: 1) document comprehension and memory population, and 2) task predictions generation using memory-enhanced inputs. In the first stage, question-context segments are passed through the RoBERTa model with LM head to obtain cross-dictionary distributions for entropy estimation. Context tokens with little independent semantic content, such as special separators, punctuation marks, and stop words, are not considered for the memory population (see details in Appendix B). Global memory is then collected from the remaining tokens from the document based on the chosen

| Model | HotpotQA, $\hat{\theta}=0.3$ | | | 2WikiMHQA, $\hat{\theta}=0.45$ | | MuSiQue, $\hat{\theta}=0.2$ | |
| | $\text{Ans}_{\pm std}$ | $\text{Supp}_{\pm std}$ | $\text{Joint}_{\pm std}$ | $\text{Ans}_{\pm std}$ | $\text{Supp}_{\pm std}$ | $\text{Ans}_{\pm std}$ | $\text{Supp}_{\pm std}$ |
|---|---|---|---|---|---|---|---|
| Task-tuned RoBERTa | $73.8_{\pm 0.44}$ | $82.86_{\pm 0.09}$ | $63.16_{\pm 0.37}$ | $65.55_{\pm 0.37}$ | $95.09_{\pm 0.02}$ | $31.46_{\pm 0.48}$ | $62.6_{\pm 0.25}$ |
| YAKE! keywords memory | $72.9_{\pm 0.29}$ | $81.67_{\pm 0.17}$ | $61.44_{\pm 0.33}$ | $63.98_{\pm 1.01}$ | $63.13_{\pm 0.27}$ | $30.30_{\pm 0.6}$ | $63.43_{\pm 0.64}$ |
| GEMFormer Highest $H$ | $73.87_{\pm 0.26}$ | $82.61_{\pm 0.42}$ | $62.85_{\pm 0.51}$ | $64.88_{\pm 0.4}$ | $94.4_{\pm 0.06}$ | $29.18_{\pm 2.39}$ | $59.7_{\pm 0.38}$ |
| GEMFormer Low ($H < \hat{\theta}$) | $\mathbf{75.13}_{\pm 0.5}$ | $\mathbf{83.8}_{\pm 0.43}$ | $\mathbf{64.77}_{\pm 0.66}$ | $\mathbf{67.14}_{\pm 0.46}$ | $\mathbf{95.67}_{\pm 0.36}$ | $31.56_{\pm 0.41}$ | $\mathbf{63.85}_{\pm 0.71}$ |
| GEMFormer Low ($H < 5^{th}\%$) | $74.19_{\pm 0.27}$ | $83.13_{\pm 0.07}$ | $63.59_{\pm 0.14}$ | $66.08_{\pm 0.39}$ | $95.15_{\pm 0.07}$ | $\mathbf{32.22}_{\pm 0.37}$ | $62.54_{\pm 0.09}$ |

Table 1: **GEMFormer with *Low H* conditions outperforms the baselines.** Table shows answer, supporting evidence, and joint F1 scores with standard deviations on dev sets (average over 3 runs) and entropy thresholds.

entropy conditions (Eq. 1). In the second stage, question and global memory tokens are concatenated with each segment for the MHQA task training. The model's weights for the first stage are updated every epoch with the weights of the model trained to solve the target task in the second stage.

We evaluated GEMFormer on three English MHQA datasets: HotpotQA distractor setting (Yang et al., 2018), 2WikiMultiHopQA (Ho et al., 2020) and MuSiQue-Ans (Trivedi et al., 2022). Further in the paper, we will refer to them as HP, 2W, and MSQ respectively. Data preprocessing, training, and inference details are described in Appendix B.

## 4 Results and Discussion

As baselines in our experiments we used a RoBERTa fine-tuned on the task without memory and a RoBERTa fine-tuned with memory bank of 200 YAKE! (Campos et al., 2020) keyword tokens from the contextual document (see Table 1). We started memory experiments from an assumption that tokens for which a language model is the most uncertain might be the most useful in the global context. However, the distribution of entropy of a task-tuned baseline model over the document tokens (see Appendix E Fig. 5) showed that the majority of the context tokens have high entropy, and locations of low entropy tokens are closely aligned with answers and supporting facts within the document. This observation points to the hypothesis that low entropy tokens might be helpful for generation because they overlap with the context with information relevant to the answer. Also, the global memory of low entropy tokens tends to guide the answer model to focus on such tokens. To ensure the correctness of this observation, we examined if the low entropy tokens are not rare entities for which the model has less amount information about. Figure 6 in Appendix E shows that the

the pre-trained model associates newly appeared tokens (<t>, </t>, [/sent] tokens) with a notably high entropy. Moreover, comparison of rare tokens entropy distributions to the overall context entropy distributions (see Table 8 in Appendix D) confirmed that rare tokens tend to have higher entropy and are infrequent in global memory. During fine-tuning, the entropy of rare tokens related to the question becomes lower compared to irrelevant ones. This allows the model to preferentially store in the memory more relevant entities. Besides, the model with a global memory filled with top-200 highest entropy tokens (*Highest H* in Equation 1 and Table 1) produced no improvement over the baselines.

We hypothesize that a model combining a tuned RoBERTa encoder with a frozen LM head for uncertainty estimation tends to detect parts of a text that are essential for the answer by reducing their uncertainty (see also Appendix B). To test this hypothesis, we used a variable-size memory consisting of tokens with entropy lower than a given threshold (*Low H* in Equation 1). The fine-tuned model with constant entropy threshold $\hat{\theta}$ outperformed the highest performing baseline memory-free model by 1.6 joint F1 points for HP, 1.59 and 0.58 points for 2W answer and supporting evidence prediction, and 1.25 F1 points for MSQ supporting paragraphs prediction. We also tested a dynamical entropy threshold as a value of the fifth percentile of the document token entropy assuming that this might help to better cover the full supporting evidence (*Low ($H < 5^{th}\%$)* in Table 1). Such a memory selection rule showed the best answer F1 for MSQ and led to a slight improvement compared to the task-tuned RoBERTa baseline but was weaker than fixed $\hat{\theta}$ for HP and 2W.

We also evaluated the ChatGPT[2] (gpt-3.5-turbo-0613 checkpoint) with memory generated

---

[2] https://openai.com/blog/chatgpt

by GEMFormer Low ($H < \hat{\theta}$). The results are presented in Table 2. We tested the question-only inputs to assess the general QA ability of the model and the two variants of input document representation (the ChatGPT prompts for our experiments are listed in Appendix C). In the first case, a doc-

| Input | HotpotQA Ans \| Sup \| Jnt | 2WikiMHQA Ans \| Sup | MuSiQue Ans \| Sup |
|---|---|---|---|
| Q | 19.89\| — \| — | 15.46\| — | 4.98\| — |
| Q+R | 24.75\|47.53\|16.43 | 25.36\|42.65 | 13.08\|40.59 |
| Q+M+R | 21.97\|43.5 \|14.84 | 26.08\|41.7 | 13.05\|38.04 |
| Q+C | 51.34\|20.6 \|11.7 | 43.22\|30.55 | 30.51\|38.6 |
| Q+M+C | 57.34\|20.9 \|12.94 | 51.27\|33.94 | 35.98\|40.33 |

Table 2: **Global explicit memory improves ChatGPT performance.** F1 scores for dev sets of HotpotQA, Musique and 2500 dev samples subset of 2WikiMHQA. Q denotes question in the model input, R – retrieved context, M – memory, C – full context document.

ument sentences were stored in a vector database and ranked by relevance to the question via the document-based QA pipeline from the `Langchain` library. The retrieved most relevant sentences were concatenated to the question and used for answer prediction. As a result, the ChatGPT performance was highly dependent on the effectiveness of the evidence retrieval process. If the retriever failed to accurately extract the evidence, the subsequent memory augmentation did not rectify this shortcoming. In the second case, we fed a model with a concatenation of a question and a contextual document with and without global memory augmentation. Although the answer F1 scores were significantly improved compared to the retrieval setting, they still fell short of the fine-tuned RoBERTa scores. However, in this setting, we observed notable improvements in answer F1 scores (+6 F1 for HP, +7.94 F1 for 2W, and +5.47 F1 for MSQ), signifying the efficacy of memory augmentation in enhancing answer generation.

**Ablation study** To verify the proposed memory augmentation, we conducted an ablation study on HP. Results are shown in Table 3. To test the importance of the question input for memory filling, we trained *No Q/Doc only* model with memory selection rule $H < 0.3$ and stage 1 entropies calculated from the document-only context. The resulting joint F1 degraded by 0.68 points indicating how question affects the memory quality. With *No fine-tune* configuration we checked if the second stage MHQA task tuning affects memory content

or general-purpose LM can be used for that as well. We tuned the model on stage 2 but froze the pretrained RoBERTa for memory uncertainty estimation. This led to the 3 joint F1 points decrease. Thus, updating the model for uncertainty estimation is critical to make memory useful. Indeed, we

| GEMFormer | Ans F1$_{\pm std}$ | Supp F1$_{\pm std}$ | Joint F1$_{\pm std}$ |
|---|---|---|---|
| Low ($H < 0.3$) | 75.13$_{\pm 0.5}$ | 83.8 $_{\pm 0.43}$ | 64.77$_{\pm 0.66}$ |
| No Q / Doc only | 74.33$_{\pm 0.16}$ | 83.67$_{\pm 0.35}$ | 64.09$_{\pm 0.28}$ |
| No fine-tune | 73.88$_{\pm 0.42}$ | 80.77$_{\pm 0.12}$ | 61.72$_{\pm 0.34}$ |
| Random memory | 72.92$_{\pm 0.21}$ | 81.76$_{\pm 0.1}$ | 61.58$_{\pm 0.25}$ |

Table 3: **Ablation study on HotpotQA dataset.** *No Q / Doc only* means excluding question from the memory population stage. *No fine-tune* uses a frozen pre-trained checkpoint for memory population and is tuned with such memory on MHQA task. *Random memory* means memory of tokens randomly chosen from the document.

measured average per token entropy during training of the GEMFormer Low ($H < 0.3$) model and found the growing difference in uncertainty between evidence and distractor facts (Fig. 2). The *Random memory* ablation was to fill memory with tokens selected randomly from the document. It showed that global memory with the arbitrary-selected content can not only fail to improve predictions but can actually degrade the model performance. We also provide the comparison of our best-

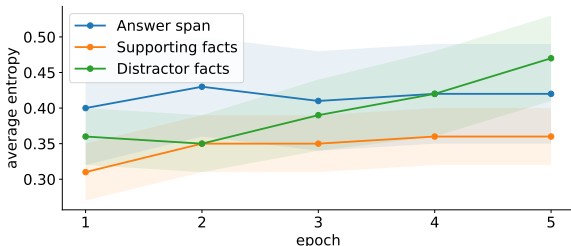

Figure 2: **Difference in uncertainty between distractor and supporting facts grows with training.** The plot shows the average per token entropy with standard deviation for answer spans, supporting evidence, and distractor facts on the HotpotQA validation set during training of the GEMFormer Low ($H < 0.3$) model.

performing GEMFormer Low ($H < \hat{\theta}$) model with existing base-sized MHQA models in Appendix A.

**Memory analysis** The results shown above demonstrate that global memory of supporting facts improves MHQA performance, and it is natural to expect that a larger memory size should give better results. We analyzed the dev set predictions of GEMFormer Low ($H < 0.3$) trained on HP with three different random seeds (Table 4) and found

that instances, where the model utilized larger average memory sizes, corresponded to higher joint F1 scores, thereby reinforcing our hypothesis. Specifically, a five-fold expansion in memory is linked to an increase of 1.32 points in the joint F1 score. Notably, runs with larger memory sizes indicate a higher coverage of evidence, albeit it does not exceed 30%. The average size of supporting facts

| GEMFormer ($H < 0.3$) joint F1 | Avg mem size$_{\pm std}$ | % of tokens from supporting facts stored in memory$_{\pm std}$ | | |
|---|---|---|---|---|
| | | total | - ans | + ans |
| 65.39 | $50_{\pm 36}$ | $30_{\pm 13}$ | $31_{\pm 13}$ | $30_{\pm 13}$ |
| 64.86 | $24_{\pm 14}$ | $21_{\pm 10}$ | $21_{\pm 10}$ | $21_{\pm 10}$ |
| 64.07 | $11_{\pm 9}$ | $10_{\pm 8}$ | $11_{\pm 8}$ | $10_{\pm 8}$ |

Table 4: **Larger average memory size and evidence coverage have higher model performance scores.** The table shows three HotpotQA runs with average memory size and the percentage of tokens from supporting facts w.r.t. supporting facts length stored in memory in total, and for samples with correct (+) and wrong (-) answers.

is 83 tokens which means that only about half of the memory is occupied by useful information with the rest of the memory slots filled with tokens from distractor paragraphs. This occupation is the same for correct and wrong answer predictions.

Counterintuitively, samples with larger memory sizes for the same model have lower answer prediction quality (Fig. 3). This could potentially be attributed to the distracting influence of irrelevant tokens stored within the memory. The larger mem-

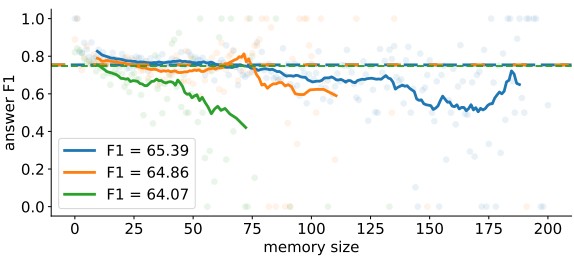

Figure 3: **Larger than average memory sizes have lower scores.** The answer F1 moving average per memory length (solid lines) and the overall answer F1 (dashed lines) for three runs of GEMFormer Low ($H < 0.3$) on HotpotQA with different joint F1.

ory size should have more unrelated tokens so the training should optimize the trade-off between enlarging memory for better evidence coverage and decreasing it to remove noise. This is supported by the results of the deeper analysis in Fig. 4. Samples with the memory containing more tokens from evidence and less irrelevant ones tend to have better

scores (Fig. 4a). And the amount of evidence stored in memory has almost no effect for low coverage (up to 30%) and leads to the answer F1 decrease for higher coverage values (Fig. 4b).

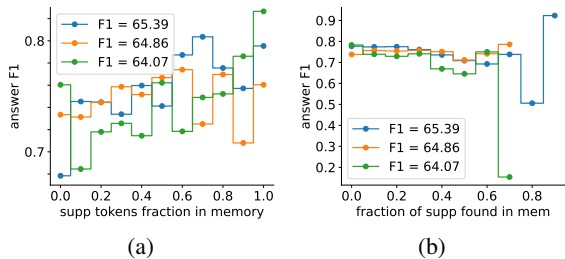

(a)                              (b)

Figure 4: **Higher occupation of memory with evidence tokens results in better scores.** Panels show dependencies of answer F1 on (a) portion of memory tokens related to supporting evidence w.r.t. *memory size*, (b) portion of memory tokens related to supporting evidence w.r.t. *supporting evidence length* for three runs of GEMFormer Low ($H < 0.3$) on HotpotQA with varying joint F1 scores.

## 5   Conclusions

In this study, we demonstrated how utilizing uncertainty-based global explicit memory can enhance the model performance on MHQA tasks. Our findings indicate that utilizing low entropy context tokens can aid the model in MHQA reasoning, but only when the entropy estimation model is specifically fine-tuned to the target task. Experiments show that higher-performing models use larger memory sizes with better coverage of supporting facts.

## Limitations

There are several limitations to this work. First, the global explicit memory augmentation of the input sequence may increase the training time by shortening the context chunk lengths. Second, the current implementation of memory token selection results in storing a significant fraction of irrelevant tokens which interferes with the calculation of correct predictions. We will work on methods to improve the relevance of information stored in memory.

## Ethics Statement

This work is a fundamental research work that focuses on technical improvement, thus we have not applied additional filtering techniques to the textual data we used, beyond what has been performed on

the original datasets. The textual data we used may have information naming or uniquely identifying individual people or offensive content that we have not been able to identify, as those are out of the focus of this work.

## Acknowledgements

AS was supported by a grant for research centers in the field of artificial intelligence, provided by the Analytical Center for the Government of the Russian Federation in accordance with the subsidy agreement (agreement identifier 000000D730321P5Q0002) and the agreement with the Moscow Institute of Physics and Technology dated November 1, 2021 No. 70-2021-00138.

Authors are thankful to SberDevices for granting access to additional computational resources.

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

# A Comparison with other MHQA methods

We also compared our best-performing GEM-Former Low $(H < \hat{\theta})$ model with existing base-sized MHQA models in Table 5. The proposed model shows the best results in MSQ answer prediction and 2W supporting evidence prediction. On HotpotQA our method achieves answer F1 comparable to ETC, BigBird, and ReadTwice and joint F1 higher than Longformer with the smaller-sized model.

# B Datasets and Training Details

All datasets' contexts are based on Wikipedia. 2WikiMultiHopQA also has additional evidence relations tuples that were not used in our experiments.

| Model | Size | HotpotQA Ans | Sup | Jnt | 2WikiMHQA Ans | Sup | MuSiQue Ans | Sup |
|---|---|---|---|---|---|---|---|---|
| GEMFormer ($H < \hat{\theta}$) | 125M | 75.13 | 83.8 | 64.77 | 67.14 | 95.67 | 31.56 | 63.85 |
| IRC (2021) | 110M | 72.9 | 79.8 | — | — | — | — | — |
| *DFGN (2019) | 110M | 69.23 | — | — | 38.49 | 57.79 | — | — |
| StepReasoner (2022) | 110M | — | — | — | 73.03 | 91.21 | — | — |
| RAG-Small (2023) | 140M | 62.8 | 49.0 | — | — | — | 24.2 | — |
| HUG-Small (2023) | 140M | 66.8 | 67.1 | — | — | — | 25.1 | — |
| SAE (2019) | 110M | 74.81 | 85.27 | 66.45 | — | — | — | — |
| Longformer-base (2020) | 149M | — | — | 64.4 | — | — | — | — |
| ETC (2020) | 166M | 75.1 | 86.9 | — | — | — | — | — |
| BigBird-ITC (2020) | 132M | 75.7 | 86.9 | 67.7 | — | — | — | — |
| BigBird-ETC (2020) | 132M | 75.5 | 87.1 | 67.8 | — | — | — | — |
| ReadTwice (2021) | 140M | 75.9 | — | — | — | — | — | — |

Table 5: **Comparison with existing methods.** Related works' F1 scores are taken from the corresponding papers. The mark * means scores taken from Fu et al. (2021).

The number of context paragraphs, the number of reasoning hops, sources, and sizes of train and dev sets for each dataset used in our experiments are presented in Table 6.

| Dataset | # para | Hops | Train | Dev |
|---|---|---|---|---|
| HotpotQA[3] | 10 | 2 | 90447 | 7405 |
| 2WikiMultiHopQA[4] | 10 | 2, 4 | 167454 | 12576 |
| MuSiQue-Ans[5] | 20 | 2-4 | 19938 | 2417 |

Table 6: **MHQA datasets statistics.**

**Preprocessing and Objective**   HotpotQA and 2WikiMultiHopQA have questions with yes/no answers and context-span answers. Also, both datasets have golden targets for supporting evidence sentences and paragraphs. We prepared HotpotQA and 2WikiMultiHopQA inputs and targets following the Longformer (Beltagy et al., 2020) MHQA approach. To prepare the input sequence, we added special tokens to indicate the question start and end, paragraph title start and end, and sentence end. The special tokens were added to the RoBERTa vocabulary and randomly initialized before fine-tuning. The training was carried out in a multi-task way to predict question types (yes/no/span), answer spans, relevant paragraphs, and evidence sentences jointly. We used the following loss function proposed by the Longformer paper authors:

$$L = \alpha_1 CE_{qtype} + \alpha_2 or\_CE_{span} +$$
$$\alpha_3 CE_{para} + \alpha_4 CE_{sent}, \quad (2)$$

where $or\_CE_{span}$ is the noisy labels handling loss function (Clark and Gardner, 2018) to account for all possible occurrences of answer span and the rest are cross-entropy losses for the classification of question types, supporting paragraphs, and evidence sentences. Weightenings $\alpha_i$ were used to keep each loss term in a similar range[6]. For HotpotQA we used $\alpha_1 = 10$, $\alpha_{2,3,4} = 1$ to reproduce the RoBERTa baseline scores from the Longformer paper. We also tested $\alpha_1$ of $1, 2$ and $5$ because of the lower values of the question type loss compared to the other loss terms during training but $\alpha_1 = 10$ showed the best results. For 2WikiMultiHopQA we did not observe the significant differences between loss terms values and used unit values for all alphas.

MuSiQue preprocessing and training were carried out following the Trivedi et al. (2022) End2End model setting. The dataset has no yes/no type of questions, only document spans. The evidence is presented with golden supporting paragraphs. So we added a paragraph start indicator token to the RoBERTa vocabulary for supporting evidence prediction. It was randomly initialized before fine-tuning. The answer span targets were selected as the last occurrence of the span in the supporting paragraphs. The multi-task training objective was a sum of an answer span cross-entropy loss $CE_{span}$ and supporting paragraphs binary cross-entropy loss $BCE_{para}$:

$$L = CE_{span} + BCE_{para}. \quad (3)$$

**Training and Hyperparameters**   For all datasets, the long input sequence was split into segments to be processed by the pre-trained RoBERTa model with 512 tokens input length. To avoid partitioning answer spans, we applied 20 token length overlaps between consecutive input chunks. The maximum memory size was limited to 200 tokens to balance the lengths of memory and context in the model input. On the memory population stage, before we collected the information from the context to the memory, we filtered out special tokens related to the supporting evidence prediction, all possible punctuation-related tokens, and stop words[7] to reduce the amount of noninformative memory content.

To start training we used the public RoBERTa-base checkpoint[8] (125M parameters) and the Hug-

[6] https://github.com/allenai/longformer/issues/143#issuecomment-733894862
[7] https://github.com/nltk/nltk/wiki/Frequently-Asked-Questions-(Stackoverflow-Edition)#how-to-remove-stopwords-with-nltk
[8] https://huggingface.co/roberta-base

| Parameter | HotpotQA | 2WikiMultiHopQA | MuSiQue |
|---|---|---|---|
| Batch size | 32 | 32 | 12 |
| Epochs | 5 | 5 | 3 |
| Optimizer | AdamW | AdamW | AdamW |
| Warmup | 1000 | 10% train steps | 1000 |
| Learning rate | 3e-5 | 3e-5 | 2e-5 |
| Compute time | 5h | 7h | 40min |

Table 7: **Training hyperparameters for all MHQA datasets**

gingface Transformers RoBERTa model implementation[9]. The HotpotQA and 2WikiMultiHopQA training and evaluation consisted of four subtasks: question type prediction using the question classification head over the first [CLS] token, answer span prediction, and evidence paragraphs and sentences prediction. The MuSiQue training and evaluation consisted of two subtasks: answer span prediction and supporting paragraphs prediction. To predict supporting evidence, we used two-layer feedforward networks with GeLU activation between layers applied over the paragraph title end and sentence end tokens for HotpotQA and 2WikiMultiHopQA and over the paragraph start tokens for MuSiQue. For the fixed entropy threshold-based memory experiments, we tested a number of threshold values and reported results with the best $\hat{\theta}$ choices to detect supporting evidence-related tokens.

During inference, the given model checkpoint and pre-trained LM head were used to generate a global memory. To compute evaluation metrics, we collected predictions across document segments and took the most probable answer as the final prediction. For HotpotQA we ensured that predicted evidence came from the two most probable paragraphs according to the dataset setup (Groeneveld et al., 2020).

Training hyperparameters are listed in Table 7. All models were trained on 8 NVIDIA A100 GPUs. The linear decay schedule without a warmup, with 1000 steps and with 10% of training steps linear warmup and constant schedule with 1000 steps warmup were tested. Linear warmup with linear decay scheduler that performed best on all datasets was used in our experiments. We also tested learning rates of 2e-5, 3e-5, and 5e-5 and epochs of 3, 5, and 7 for hyperparameter search.

**Uncertainty Estimation Procedure** GEM-Former stage 1 includes predicting token

distributions with the task-tuned RoBERTa encoder followed by a pre-trained LM head. By this, we combined the reasoning skills of the tuned encoder and the language structure knowledge of the LM head. The RoBERTa pre-training data contains Wikipedia corpus (Liu et al., 2019) and the datasets we used are also based on Wikipedia. This assures that pre-trained RoBERTa LM head weights possess information about the language distribution of the context documents in the training set. The frozen LM head is able to map the output representations of the encoder while preserving the uncertainty of predictions. As the encoder is progressively fine-tuned for MHQA task, its output representations become more certain for tokens related to the answer and supporting facts. During our primary experiments, we also tested the LM head jointly fine-tuned with the encoder model. It led to the same results as a baseline and did not yield any significant differences in uncertainty values for answer and supporting facts compared to other parts of the text. This observation can be attributed to the inherent nature of the LM task, which aims to predict all tokens in a sequence without any specific focus on the MHQA reasoning.

## C ChatGPT evaluation prompts

The prompt for experiments with the retrieved context was the following: System: You are a world-class algorithm to answer questions in a specific format. Human: Answer the question using the following context <context>. Question: <question>. Tips: Make sure to answer in the correct format.

To combine the question and the full contextual document in the model input (*Question+Context* in Table 2) we used the following prompt: System: You are a world-class algorithm to answer questions based on the given text. Store the answer and the supporting evidence from the text in the following structure: {'answer': 'answer string', 'supporting evidence': ['list of supporting evidence sentences']}. Human: Text: <context> Question: <question>.

Finally, the memory-augmented input (*Question+Memory+Context* in Table 2) prompt was the following: System: You are a world-class algorithm to answer questions based on the given text and memory.

---

[9]https://huggingface.co/docs/transformers/model_doc/roberta

```
Use the provided memory to detect
question-related information in text.
Store the answer and the supporting
evidence from the text in the following
structure: {'answer': 'answer string',
'supporting evidence': ['list of
supporting evidence sentences']}. Human:
Memory: <mem> Text: <context> Question:
<question>.
```

## D    Rare tokens entropy

We collected rare (occurring in a context less than 5 times) tokens from each contextual document of the validation set for each dataset to compare the characteristics of the entropy distributions of rare tokens to the overall context entropy distributions. The descriptive statistics averaged over the validation set samples are presented in Table 8. As a result, we can see that the mean, median, mode and maximum values of rare tokens distributions match overall context distributions up to two decimal places for all three datasets. Also, both distribution types have skewness coefficients lower than $-1$ for each dataset, which means the distributions are skewed to the left. This observation confirms that rare tokens most frequently have high entropy values. We also calculated how many global memory tokens are rare: for HotpotQA $61 \pm 21\%$ of memory tokens are rare ones, for 2WikiMHQA – $9 \pm 9\%$, and for MuSiQue – $16 \pm 15\%$. These observations imply that, in practice, rare tokens tend to have high entropy relative to the overall context entropy, and for two out of three datasets rare entities are infrequent in global memory. The rare tokens analysis suggests that during fine-tuning, the entropy of question-related rare tokens decreases compared to the entropy of irrelevant tokens.

## E    Entropy over document distribution heatmaps

In this section, we listed per token entropy heatmaps for one validation context example. Heatmaps illustrate that the pre-trained model's uncertainty is almost uniformly low except for newly-added tokens (see Fig. 6). Looking at the entropy difference after the first epoch and before fine-tuning, we can see how the uncertainty of title and sentence indicator tokens decreases, and the supporting facts have tokens with unchanged uncertainty, while the entropy of the rest of the document increases (see Fig. 7).

| Dataset | Tokens | Min | Mean | Median | Mode | Max | Skewness | Kurtosis |
|---------|--------|-----|------|--------|------|-----|----------|----------|
| HotpotQA | Rare | $0.043_{\pm 0.040}$ | $0.465_{\pm 0.034}$ | $0.487_{\pm 0.039}$ | $0.495_{\pm 0.065}$ | $0.621_{\pm 0.017}$ | $-1.30_{\pm 0.56}$ | $2.42_{\pm 2.66}$ |
| | Overall | $0.017_{\pm 0.021}$ | $0.460_{\pm 0.038}$ | $0.487_{\pm 0.042}$ | $0.489_{\pm 0.076}$ | $0.625_{\pm 0.016}$ | $-1.38_{\pm 0.59}$ | $2.51_{\pm 2.72}$ |
| 2WikiMHQA | Rare | $0.101_{\pm 0.065}$ | $0.503_{\pm 0.025}$ | $0.530_{\pm 0.026}$ | $0.556_{\pm 0.033}$ | $0.628_{\pm 0.015}$ | $-1.47_{\pm 0.48}$ | $2.73_{\pm 2.58}$ |
| | Overall | $0.081_{\pm 0.059}$ | $0.503_{\pm 0.025}$ | $0.529_{\pm 0.027}$ | $0.553_{\pm 0.045}$ | $0.635_{\pm 0.016}$ | $-1.42_{\pm 0.45}$ | $2.51_{\pm 2.40}$ |
| MuSiQue | Rare | $0.006_{\pm 0.012}$ | $0.410_{\pm 0.012}$ | $0.430_{\pm 0.012}$ | $0.450_{\pm 0.013}$ | $0.586_{\pm 0.015}$ | $-1.55_{\pm 0.23}$ | $3.33_{\pm 1.09}$ |
| | Overall | $0.003_{\pm 0.006}$ | $0.409_{\pm 0.014}$ | $0.431_{\pm 0.013}$ | $0.452_{\pm 0.013}$ | $0.590_{\pm 0.015}$ | $-1.59_{\pm 0.23}$ | $3.37_{\pm 1.26}$ |

Table 8: **Descriptive statistics for all context tokens' distribution and rare tokens distribution.** The values are averaged over the validation set samples for HotpotQA, 2WikiMHQA, and MuSiQue datasets. Skewness is calculated with Fisher-Pearson coefficient and for kurtosis calculation the Fisher's definition was used.

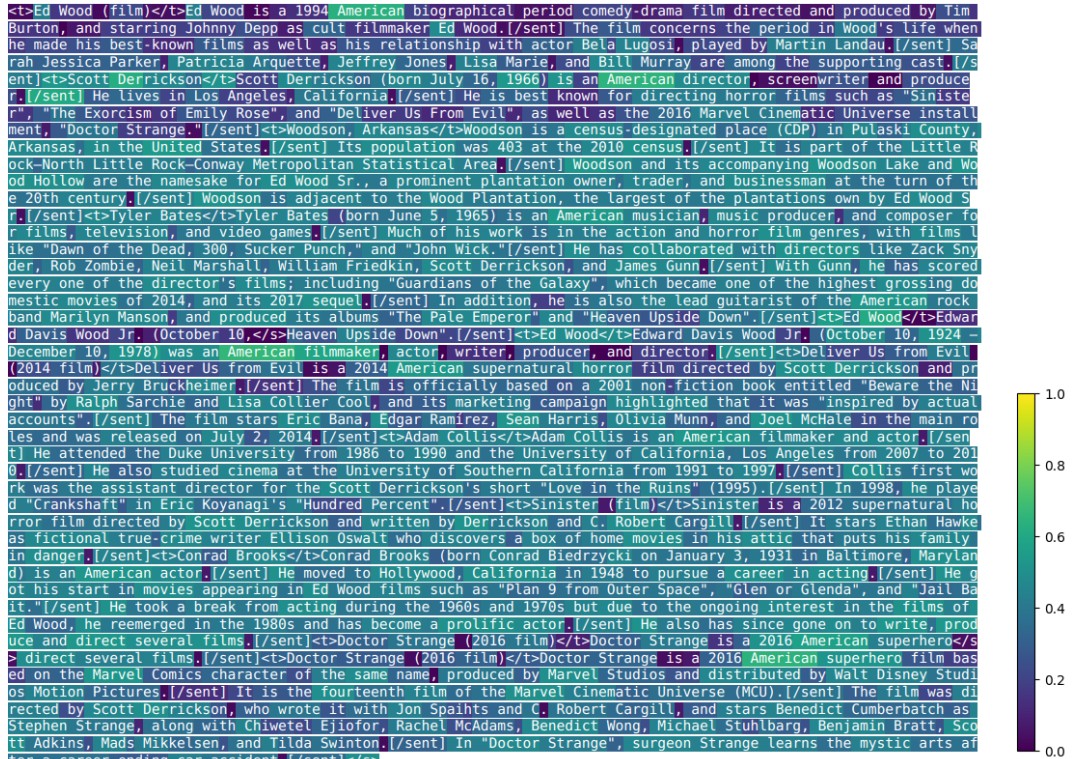

Figure 5: **Supporting facts appear in the areas of the low entropy of the context.** The fine-tuned RoBERTa baseline entropy distribution heatmap. The presented text is a sample context from the HotpotQA validation set with the following supporting evidence: *"Scott Derrickson (born July 16, 1966) is an American director, screenwriter and producer.", "Edward Davis Wood Jr. (October 10, 1924 – December 10, 1978) was an American filmmaker, actor, writer, producer, and director."*

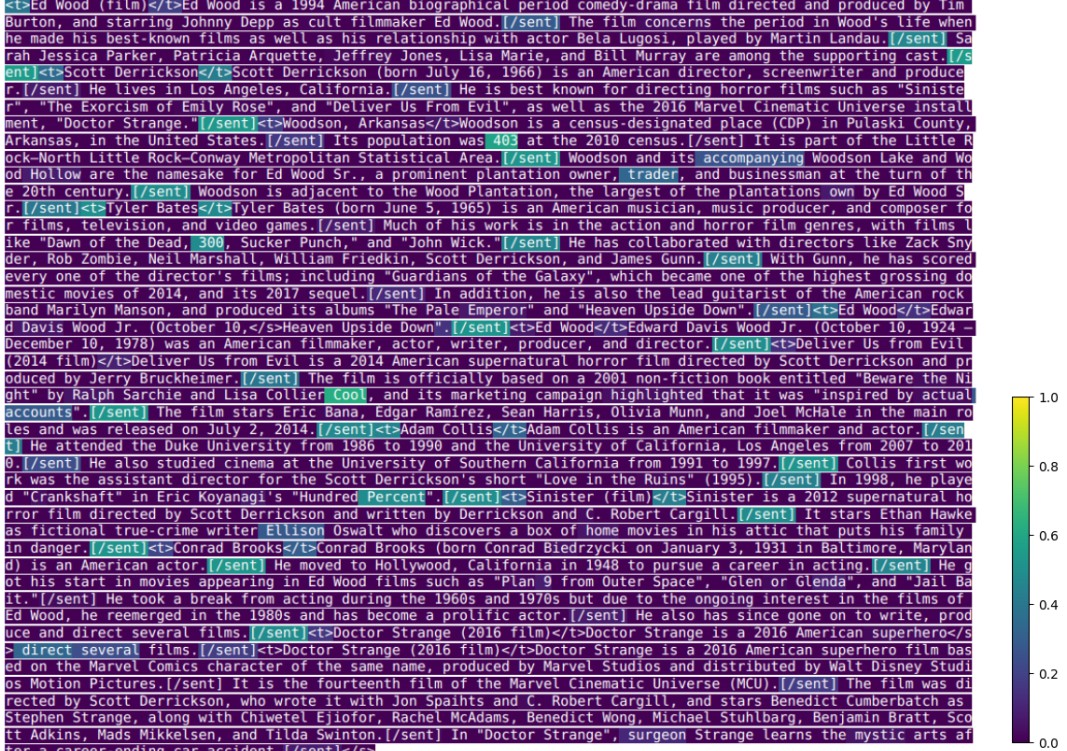

Figure 6: **The pre-trained model has uniformly low uncertainty about context except for sentence and paragraph marker tokens, added to the vocabulary before fine-tuning.** The pre-trained RoBERTa entropy heatmap of an example from the HotpotQA validation set.

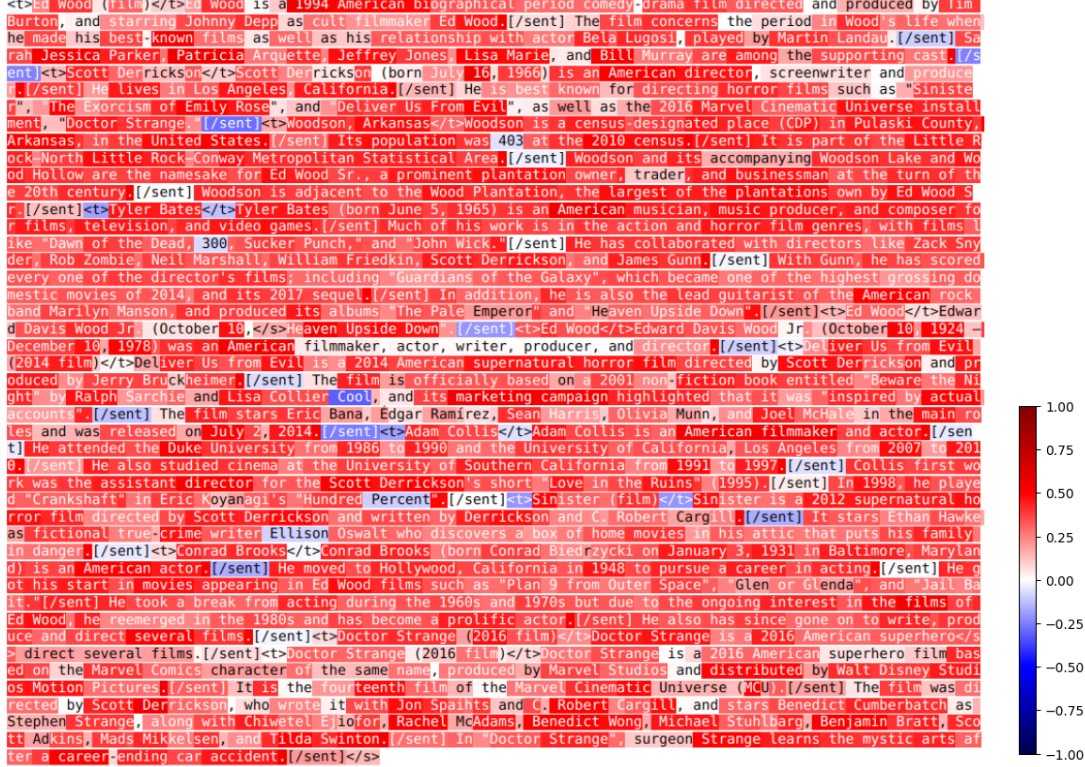

Figure 7: **After one fine-tuning epoch with GEMFormer Low (H < 0.3), supporting facts tokens include elements with unchanged uncertainty values, while the entropy of the rest of the document has increased.** The HotpotQA validation set example heatmap depicting the difference in entropy values after the first epoch of fine-tuning and before fine-tuning.