# OpenReview forum: "Uncertainty Guided Global Memory Improves Multi-Hop Question Answering"
_EMNLP/2023/Conference — EMNLP 2023 Main_

### Official Review · Reviewer_E7Ta · 2023-07-25

**Soundness:** 3

**Excitement:**

3: Ambivalent: It has merits (e.g., it reports state-of-the-art results, the idea is nice), but there are key weaknesses (e.g., it describes incremental work), and it can significantly benefit from another round of revision. However, I won't object to accepting it if my co-reviewers champion it.

**Paper Topic And Main Contributions:**

The authors present a novel method for solving the multi-hop question answering task, namely GEMFormer, which is a pre-trained model that is fine-tuned with memory augmented input. The results are improved over the baseline on three datasets.

**Questions For The Authors:**

Line 063: Is there previous literature on defining token importance as language model uncertainty? Some more background on this idea would be handy in the related work section.

**Reasons To Accept:**

The paper presents an interesting new model for memory augmentation.

**Reasons To Reject:**

The paper is not clearly structured. For instance, a lot of content in the Results and Discussion section describes the methods used which have not yet been discussed, making it difficult to follow.

**Reproducibility:**

3: Could reproduce the results with some difficulty. The settings of parameters are underspecified or subjectively determined; the training/evaluation data are not widely available.

**Reviewer Confidence:**

1: Not my area, or paper was hard for me to understand. My evaluation is just an educated guess.

**Typos Grammar Style And Presentation Improvements:**

Line 006: The sentence is too long and not clearly punctuated

Line 093: Same comment as line 006, too many conjuncts in a sentence make it difficult to read

---

> ### Author Rebuttal · Authors · 2023-08-28
>
> **We are thankful for your feedback and thoughtful examination of our paper. Your comments provide valuable guidance for improving the clarity and overall appeal of our work.**
>
> **Addressing Clarity in the Results and Discussion Section:**
>
> We genuinely thank you for raising this concern. The Results and Discussion section indeed encompasses a range of evaluations, including experimental results, ablation studies, and analyses of the impact of memory content on model performance. We grouped these assessments within a single section due to space constraints. The intention behind this approach is to present a comprehensive overview of our methodology's effectiveness. To enhance clarity, we will ensure that the motivations behind the ablation studies and analysis methods are clearly articulated in the camera-ready version of the paper.
>
> **Responses to Queries:**
>
> Query: Line 063: Is there previous literature on defining token importance as language model uncertainty? Some more background on this idea would be handy in the related work section.
> Response: Similar to our usage of language model predictions, in [1, 2] authors use a language model (LM) probability in machine translation task.
> A machine translation model probability and LM probability are used to measure the pointwise mutual information (MI) between each target token and its source context under the condition of the previous target context. The estimated values are considered token importance and are used as weightings to rescale the training loss. Such a training approach makes the model pay more attention to more “informative" tokens.
> To the best of our knowledge, there is no literature specifically discussing token importance based on language model uncertainty. Nonetheless, our motivation behind this approach was the following. Language model entropy for a given token is a conditional entropy with respect to the question and the contextual document. It shows the amount of information needed to describe the token given the question with the task-defining information and the source context. So lower uncertainty values should reflect tokens that are more relevant to the task.
>
> **Refinements in Language and Presentation:**
>
> We sincerely appreciate your observation regarding sentence length and punctuation. We have revised the sentences in question as follows:
>
> Line 006: The landscape of MHQA approaches can be classified into two primary categories. The first group focuses on extracting supporting evidence, thereby constraining the QA model's context to predicted facts. Conversely, the second group relies on the attention mechanism of the long input encoding model to facilitate multi-hop reasoning.
>
> Line 093: GEMFormer adapts the idea of interpretable memory by storing tokens from the input sequence. This augmentation of memory enriches long inputs with essential global contextual information important for the task.
>
> **Low Reproducibility Score:**
>
> We understand the importance of reproducibility in research findings. As part of our code release, we will include detailed explanations of how the global memory was populated and how the baseline and the models with memory augmentation were fine-tuned, allowing fellow researchers to adapt our approach to the new QA and multi-hop QA datasets. The datasets we used are all publicly available. We are committed to ensuring that our work is as reproducible as possible. If you have specific concerns about reproducibility, we would be happy to address them.
>
>
> Once again, we thank you for your invaluable feedback, and we are committed to refining our paper in alignment with your suggestions.
>
>
> [1] Songming Zhang, Yijin Liu, Fandong Meng, Yufeng Chen, Jinan Xu, Jian Liu, and Jie Zhou. 2022. Conditional Bilingual Mutual Information Based Adaptive Training for Neural Machine Translation. In Proceedings of the 60th Annual Meeting of the Association for Computational Linguistics (Volume 1: Long Papers), pages 2377–2389, Dublin, Ireland. Association for Computational Linguistics.
>
> [2] Aditi Jain, Nishant Kambhatla, and Anoop Sarkar. 2023. Language Model Based Target Token Importance Rescaling for Simultaneous Neural Machine Translation. In Proceedings of the 20th International Conference on Spoken Language Translation (IWSLT 2023), pages 341–356, Toronto, Canada (in-person and online). Association for Computational Linguistics.

---

### Official Review · Reviewer_Wfje · 2023-08-04

**Soundness:** 2

**Excitement:**

2: Mediocre: This paper makes marginal contributions (vs non-contemporaneous work), so I would rather not see it in the conference.

**Missing References:**

Please refer to quite some models listed in: https://arxiv.org/abs/2204.09140

**Paper Topic And Main Contributions:**

The paper proposed a two-stage transformer-based framework GEMFormer towards multi-hop question answering task. In specific, GEMFormer first generate global memory based on given question and multiple document segments, then reuse question, document segments and newly produced global memory to generate answers. In first stage, the global memory is populated by ranking an entropy value over all tokens in the input sequence.

**Questions For The Authors:**

A: How to define uncertainty threshold on new multi-hop dataset?
B: Is GEMFormer better than other multi-hop QA models? (https://arxiv.org/abs/2204.09140)

**Reasons To Accept:**

The proposed global memory population mechanism is indeed an evidence collection mechanism with heuristic threshold rules, which requires no additional training.

**Reasons To Reject:**

(1) The heuristic uncertainty threshold used in first-stage of memory population requires specific values for different dataset, which is lack of generalization and reproducibility.
(2) The overall framework perhaps works on general long document question answering and more LLMs. The authors only use Roberta and multi-hop QA for testing. In contrast, if the authors believe they made special progress on multi-hop QA, then analysis on hop-depend questions should be provided. For example, how GEMFormer is superior than other previous LLMs that can deal with multi-hop QA as well. There're only basic Roberta model be listed as a baseline.

**Reproducibility:**

3: Could reproduce the results with some difficulty. The settings of parameters are underspecified or subjectively determined; the training/evaluation data are not widely available.

**Reviewer Confidence:**

4: Quite sure. I tried to check the important points carefully. It's unlikely, though conceivable, that I missed something that should affect my ratings.

---

> ### Author Rebuttal · Authors · 2023-08-28
>
> **We sincerely appreciate your valuable feedback and constructive insights. Below, we address each of your points.**
>
> **Addressing the Dataset-Specific Uncertainty Threshold Values:**
>
> We acknowledge your concern regarding the dataset-specific uncertainty threshold values used in the first stage of the memory population. In this regard, our method is not different from any other method of fine-tuning or trainable model augmentation (i.e. LoRA) that assumes training on the target dataset. Designing a method that can generalize to other tasks without fine-tuning is not the goal of the presented study and achieving such generalization is not claimed in the text. This places GEMFormer in the research area where pre-trained LMs are augmented and fine-tuned for specific tasks to achieve better quality compared to more universal models. We believe that the research in this area still has significant value for the NLP community and deserves publication.
>
> We understand the importance of reproducibility in research findings. As part of our code release, we will include detailed explanations of how these thresholds were chosen, allowing fellow researchers to adapt our approach to the new QA and multi-hop QA datasets.  The datasets we used are all publicly available. We are committed to ensuring that our work is as reproducible as possible. If you have specific concerns about reproducibility, we would be happy to address them.
>
> **Clarifications on the Roberta Usage and the Comparison with Previous MHQA LMs:**
>
> The pre-trained RoBERTa-base was selected as a model starting point following the Longformer, BigBird, and ETC architectures that represent base-sized generative Transformers from the top of the HotpotQA leaderboard to the moment of the manuscript submission. We compare the performance of our approach not only to basic RoBERTa but also to other relevant MHQA models such as Longformer, ETC, BigBird, and ReadTwice. Please, refer to Table 4 in Appendix A for a comparison with 11 other models. This table underscores the improvements our approach brings to this specific task.
>
>  We also include in comparison GPT-3.5 as a universal generalizable LLM alternative for answering multi-hop questions. ChatGPT evaluation results can be found in Table 7 of Appendix C and the extended data are provided in the following Table 1.
>
>
> **Answers to Questions:**
>
> Question A: How to define uncertainty threshold on new multi-hop dataset?
>
> Answer: In our experiments, we conducted a systematic grid search within a range from the minimal value to the mode (not included) of entropy distributions averaged across train samples. We used the fine-tuned baseline entropy values. The optimal threshold value for memory augmentation was determined through experimentation. This approach ensures that the threshold value is adaptable to new multi-hop datasets while maintaining the performance gains achieved through our method.
>
> Question B: Is GEMFormer better than other multi-hop QA models? (https://arxiv.org/abs/2204.09140)
>
> Answer: The comparison between GEMFormer and other multi-hop QA models is thoroughly addressed in Appendix A (Table 4) of our paper. DFGN, SAE, and ETC architectures mentioned in the suggested review paper are presented in Table 4.
>
> Notably, we have also extended our evaluation to include GPT-3.5, as detailed in Appendix C (Table 7) and in the revised version of our GPT-3.5 experimental results in  Table 1 below.
>
> | ChatGPT Input                         || HotpotQA || 2WikiMHQA || MuSiQue||
> |---------------------------------------|:---------:|:--------:|:--------:|:---------:|:-------:|:---------:|:-------:|
> |                                       | Ans F1 |  Supp F1 | Joint F1 | Ans F1 | Supp F1 | Ans F1 | Supp F1 |
> | Question                              |   $19.89$   |     -    |     -    |   $15.46$   |    -    |    $4.98$   |    -    |
> | Question + Retrieved Context          |   $24.75$   |   $47.53$  |  $ 16.43$  |   $25.36$   |  $42.65  |  $ 13.08 $  |  $40.59$  |
> | Question + Memory + Retrieved Context |   $21.97$   |  $ 43.5 $  |   $14.84$  |  $ 26.08 $  |  $ 41.7$  |   $13.05$   |  $38.04$  |
> | Question + Context                    |  $ 51.34$   |  $ 20.6 $  |   $11.7$   |   $43.22$   |  $30.55$  |  $ 30.51 $  |  $ 38.6$  |
> | Question + Memory + Context           |   $57.34$   |   $20.9$   |   $12.94$  |   $51.27$   |  $33.94$  |   $35.98$   |  $40.33$  |
>
> Table 1. F1 scores for dev sets of HotpotQA, Musique, and 2500 samples subset of 2WikiMHQA subset for GPT-3.5-turbo checkpoint.
>
>
> We would like to thank you for your review once again. We hope that our clarifications and additional data clarify the contribution and significance of our study for QA tasks. We are confident that the revisions made in response to your feedback will strengthen the quality of our submission.

---

### Official Review · Reviewer_LDqA · 2023-08-12

**Soundness:** 3

**Excitement:**

3: Ambivalent: It has merits (e.g., it reports state-of-the-art results, the idea is nice), but there are key weaknesses (e.g., it describes incremental work), and it can significantly benefit from another round of revision. However, I won't object to accepting it if my co-reviewers champion it.

**Paper Topic And Main Contributions:**

The paper focus on multi-hop question answering task. A new way of using entropy values of entities to extract a set of tokens named as global memory has been presented in this paper. It computes the entropy for each token from the probabilities of tokens returned by RoBERTa language modeling head layer.  Afterward, the constructed memory is used along with given context to generate answer for question.

**Questions For The Authors:**

1. It would be interesting to see whether the answer model really focuses on the GM tokens or it inversely guide model not to focus on them. Entropy will be lower when model have less amount information about a token, which implies that rare entities should have lower entropy. It is not clear those rare entities are really important to generate target answer.
2. If entropy value based extraction method can identify the important tokens, then why GPT3.5 model performance degrades.

**Reasons To Accept:**

Entropy-guided token extraction is a novel way to build a global memory.

**Reasons To Reject:**

1. It is not clear that how tokens with low entropy can be important to generate answer. It needs elaborate discussion on what type of tokens are being extracted in the global memory.
2. Important baseline is missing. There is no comparision with very obvious method of building memory bank using existing keywords extraction methods as the main purpose of this paper is preparing a set of tokens that would be helpful for answer. Without further experimental results, it is hard to evaluate the significance of this method.

**Reproducibility:**

5: Could easily reproduce the results.

**Reviewer Confidence:**

4: Quite sure. I tried to check the important points carefully. It's unlikely, though conceivable, that I missed something that should affect my ratings.

---

> ### Author Rebuttal · Authors · 2023-08-28
>
> **We would like to express our gratitude for your in-depth review of our paper. We have carefully considered each of your comments and concerns.**
>
> **Addressing the Importance of Low Entropy Tokens for Answer Generation:**
>
> We have included an entropy heatmap as Figure 5 in Appendix D to visually demonstrate that the locations of low entropy tokens are closely aligned with supporting facts within the document. This observation points to the hypothesis that low entropy tokens might be helpful for generation because they overlap with the context with information relevant to the answer. This is supported by the analysis showing that performance is positively affected by the increased fraction of memory tokens retrieved from evidence (see Figure 4a in the main text). Thus, our current understanding here is that the inclusion of low entropy tokens in the global memory serves to increase the model's attention towards evidence sentences, facilitating a more informed answer generation process.
> Also, the general motivation behind the entropy-based memory population is the following. We feed the model with a question and a context and compute entropy for each contextual token. This entropy is conditional with respect to the question and token-surrounding context. Such entropy measures how much entropy a token has remaining if we have already learned the question and the context. In other words, how easily the token can be predicted given the question and the context. Taking into account that a document is a collection of question-relevant and distractor paragraphs, the entropy of a task-relevant token should be lower than the entropy of the irrelevant one.
>
>
> **Addressing the Baseline with a Memory Bank Consisting of Extracted Keywords:**
>
> Following your suggestion to evaluate GEMFormer against a simple but relevant baseline, we have performed experiments using the YAKE! [1] method for the extraction of keywords from the context to populate a memory bank of 200 tokens for each data sample. The results (see Table 1 below) consistently show that the model's performance with the GEMFormer's memory augmentation (Table 1 of the main text) surpasses that achieved by the memory bank approach using extracted keywords. This underlines the effectiveness of our proposed method in capturing and utilizing relevant information for improved question answering. Moreover, keywords in memory lead to a performance inferior to the memoryless RoBERTa baseline.
>
> | Dataset   | Ans F1$_{\pm std}$ | Supp F1$_{\pm std}$ | Joint F1$_{\pm std}$ |
> |-----------|:---------------------:|:-------------------:|:--------------------:|
> | HotpotQA  |     $72.9_{\pm 0.29}$    |   $81.67_{\pm 0.17}$   |     $61.44_{\pm 0.33} $   |
> | 2WikiMHQA |    $63.98_{\pm 1.01}$    |   $63.13_{\pm 0.27}$   |           -          |
> | MuSiQue   |     $30.30_{\pm 0.6}$    |   $63.43_{\pm 0.64}$   |           -          |
> Table 1. Results of fine-tuning Roberta with global memory of YAKE! keywords.
>
> **Answers to Questions:**
>
> Question 1:  It would be interesting to see whether the answer model really focuses on the GM tokens or it inversely guide model not to focus on them. Entropy will be lower when model has less amount information about a token, which implies that rare entities should have lower entropy. It is not clear those rare entities are really important to generate target answer.
>
> Answer: 1) The observation you made regarding the lower entropy of rare entities implying their importance is astute. However, upon examining Figure 6 in Appendix D, we find that the entropy of the pre-trained model associated with special tokens such as <t>, </t>, and [/sent] (which are new to the model's vocabulary) is notably high. This observation indicates that the model lacks information about special tokens, contrary to the expectation that rare entities should have low entropy. Consequently, our model indeed concentrates on GM tokens, and the inclusion of such tokens in the memory serves to enhance the model's focus on supporting facts.
> 2) We collected rare (occurring in a context less than 5 times) tokens from each contextual document of the validation set for each dataset to compare the characteristics of the entropy distributions of rare tokens to the overall context entropy distributions. The descriptive statistics averaged over the validation set samples are presented in Table 2. As a result, we can see that the mean, median, mode and maximum values of rare tokens distributions match overall context distributions up to two decimal places for all three datasets. Also, both distribution types have skewness coefficients lower than $-1$ for each dataset, which means the distributions are skewed to the left. This observation confirms that rare tokens most frequently have high entropy values.
> We also calculated how many global memory tokens are rare: for HotpotQA $61\pm 21\\%$ of memory tokens are rare ones, for 2WikiMHQA – $9\pm 9\\%,$ and for MuSiQue – $16\pm 15 \\%$. These observations imply that, in practice, rare tokens tend to have high entropy relative to the overall context entropy, and for two out of three datasets rare entities are infrequent in global memory. The rare tokens analysis suggests that during fine-tuning, the entropy of rare tokens related to the question becomes lower compared to irrelevant ones. This allows the model to preferentially store in the memory more relevant entities.
>
> |    | Tokens            |        Min       |       Mean       |      Median      |       Mode       |        Max       | Skewness (Fisher-Pearson coef.) | Kurtosis (Fisher) |
> |-----------|-----------------------|:----------------:|:----------------:|:----------------:|:----------------:|:----------------:|:-------------------------------:|:-----------------:|
> | HotpotQA  | Rare | $0.043_{\pm 0.040}$ | $0.465_{\pm 0.034}$ | $0.487_{\pm 0.039}$ | $0.495_{\pm 0.065}$ | $0.621_{\pm 0.017}$ |         $-1.30_{\pm 0.56}$         |   $2.42_{\pm 2.66}$  |
> |           | Overall          | $0.017_{\pm 0.021}$ | $0.460_{\pm 0.038}$ | $0.487_{\pm 0.042}$ | $0.489_{\pm 0.076}$ | $0.625_{\pm 0.016}$ |         $-1.38_{\pm 0.59}$         |   $2.51_{\pm 2.72}$  |
> | 2WikiMHQA | Rare | $0.101_{\pm 0.065}$ | $0.503_{\pm 0.025}$ | $0.530_{\pm 0.026}$ | $0.556_{\pm 0.033}$ | $0.628_{\pm 0.015}$ |         $-1.47_{\pm 0.48}$         |   $2.73_{\pm 2.58}$  |
> |           | Overall         | $0.081_{\pm 0.059}$ | $0.503_{\pm 0.025}$ | $0.529_{\pm 0.027}$ | $0.553_{\pm 0.045}$ | $0.635_{\pm 0.016}$ |         $-1.42_{\pm 0.45}$         |   $2.51_{\pm 2.40}$  |
> | MuSiQue   | Rare | $0.006_{\pm 0.012}$ | $0.410_{\pm 0.012}$ | $0.430_{\pm 0.012}$ | $0.450_{\pm 0.013}$ | $0.586_{\pm 0.015}$ |         $-1.55_{\pm 0.23}$         |   $3.33_{\pm 1.09}$  |
> |           | Overall          | $0.003_{\pm 0.006}$ | $0.409_{\pm 0.014}$ | $0.431_{\pm 0.013}$ | $0.452_{\pm 0.013}$ | $0.590_{\pm 0.015}$ |         $-1.59_{\pm 0.23}$         |   $3.37_{\pm 1.26}$  |
>
> Table 2. Descriptive statistics for all context tokens’ distribution and rare (less than 5 occurrences in the context document) tokens distribution averaged over the validation set samples (with standard deviation values) for HotpotQA, 2WikiMHQA, and MuSiQue datasets.
>
> Question 2: If entropy value based extraction method can identify the important tokens, then why GPT3.5 model performance degrades.
>
> Answer: GPT-3.5 scores provided in the paper were obtained following the common practice of storing a contextual document in a vector database as a collection of document sentences and retrieving sentences by relevance to the question. In experiments, we used the document-based QA pipeline from the Langchain library. The most relevant sentences were concatenated to the question and sent to ChatGPT for answer prediction. In this case, we found that the final performance of GPT-3.5 is highly dependent on the effectiveness of the evidence retrieval process. If the retriever fails to accurately extract evidence, the subsequent memory augmentation does not rectify this shortcoming. Furthermore, we conducted additional experiments where we fed a model with a concatenation of a question and a contextual document with and without global memory augmentation. The detailed results are presented in Table 3.
>
> | ChatGPT Input                         || HotpotQA || 2WikiMHQA || MuSiQue||
> |---------------------------------------|:---------:|:--------:|:--------:|:---------:|:-------:|:---------:|:-------:|
> |                                       | Ans F1 |  Supp F1 | Joint F1 | Ans F1 | Supp F1 | Ans F1 | Supp F1 |
> | Question                              |   $19.89$   |     -    |     -    |   $15.46$   |    -    |    $4.98$   |    -    |
> | Question + Retrieved Context          |   $24.75$   |   $47.53$  |  $ 16.43$  |   $25.36$   |  $42.65  |  $ 13.08 $  |  $40.59$  |
> | Question + Memory + Retrieved Context |   $21.97$   |  $ 43.5 $  |   $14.84$  |  $ 26.08 $  |  $ 41.7$  |   $13.05$   |  $38.04$  |
> | Question + Context                    |  $ 51.34$   |  $ 20.6 $  |   $11.7$   |   $43.22$   |  $30.55$  |  $ 30.51 $  |  $ 38.6$  |
> | Question + Memory + Context           |   $57.34$   |   $20.9$   |   $12.94$  |   $51.27$   |  $33.94$  |   $35.98$   |  $40.33$  |
>
> Table 3. F1 scores for dev sets of HotpotQA, Musique, and 2500 samples subset of 2WikiMHQA subset for GPT-3.5-turbo checkpoint.
>
> The prompt for question+context inputs was the following:
>
> *System: You are a world-class algorithm to answer questions based on the given text. Store the answer and the supporting evidence from the text in the following structure: {‘answer’: ‘answer string’, supporting evidence’: [“list of supporting evidence sentences”]
> Human: Text: <context> Question: <question>*
>
> The memory-augmented input prompt was the following:
>
> *System: You are a world-class algorithm to answer questions based on the given text and memory. Use the provided memory to detect question-related information in text. Store the answer and the supporting evidence from the text in the following structure: {‘answer’: ‘answer string’, ‘supporting evidence’: [“list of supporting evidence sentences”]}
> Human: Memory: <mem> Text: <context> Question: <question>*
>
> Although the answer prediction scores were significantly improved compared to the retrieval setting of GPT-3.5, they still fell short of the fine-tuned RoBERTa scores. However, in this setting, we observed notable improvements in answer F1 scores ($+6$ F1 for HotpotQA, $+7.94$ F1 for 2WikiMHQA, and $+5.47$ F1 for MuSiQue), signifying the efficacy of memory augmentation in enhancing answer generation.
>
> We greatly appreciate your attention to these critical aspects of our work. Your evaluation and questions have guided us to provide more comprehensive explanations and supporting evidence for our manuscript. We are confident that these clarifications further underscore the significance and validity of our proposed GEMFormer method for multi-hop question answering. We hope that our efforts in addressing your concerns will lead to a more compelling paper that merits acceptance to EMNLP.
>
> Thank you once again for your time and invaluable feedback.
>
> [1] Campos R., Mangaravite V., Pasquali A., Jorge A.M., Nunes C., and Jatowt A. (2018). A Text Feature Based Automatic Keyword Extraction Method for Single Documents. In: Pasi G., Piwowarski B., Azzopardi L., Hanbury A. (eds). Advances in Information Retrieval. ECIR 2018 (Grenoble, France. March 26 – 29). Lecture Notes in Computer Science, vol 10772, pp. 684 - 691.

---

### Official Review · Reviewer_yjSH · 2023-08-12

**Soundness:** 4

**Excitement:**

4: Strong: This paper deepens the understanding of some phenomenon or lowers the barriers to an existing research direction.

**Paper Topic And Main Contributions:**

This paper introduces GEMFormer (Global Explicit Memory Transformer), a method to augment the input to a pre-trained transformer model like RoBERTa with a global memory sequence consisting of tokens from the document that are important for answering the question. This leads to improvement in multi-hop question answering (MHQA) which involves reasoning over multiple parts of a long document to answer a question. The method has two stages - First, the uncertainty of each token is estimated using the pre-trained language model head of RoBERTa. Counterintuitively, tokens with low uncertainty are found to be important and are added to the memory. In the second stage, the memory-augmented input is fed to RoBERTa to make final MHQA predictions. Experiments on three MHQA datasets show improvements over the RoBERTa baseline.

**Questions For The Authors:**




**Reasons To Accept:**

The paper is well-written and easy to follow. The proposed GEMFormer method consistently surpasses the RoBERTa baselines across three MHQA datasets. Extensive ablation studies analyze the impact of the question context, fine-tuning, and memory content on performance, providing insights into why the proposed approach works.


**Reasons To Reject:**

While results are consistently better than the baseline, absolute improvements are marginal - 1-3 F1 points over a strong RoBERTa baseline.


**Reproducibility:**

4: Could mostly reproduce the results, but there may be some variation because of sample variance or minor variations in their interpretation of the protocol or method.

**Reviewer Confidence:**

3: Pretty sure, but there's a chance I missed something. Although I have a good feel for this area in general, I did not carefully check the paper's details, e.g., the math, experimental design, or novelty.

---

> ### Author Rebuttal · Authors · 2023-08-28
>
> **We sincerely appreciate your thoughtful feedback and constructive critique of our paper.**
>
> **Clarifications on Marginal Improvements:**
>
> We acknowledge your concern about the marginal absolute improvements over the strong RoBERTa baseline. The complex nature of MHQA tasks, which require reasoning across diverse parts of lengthy documents, often poses challenges for even incremental enhancements. GEMFormer improvement in performance is modest but consistent. It outperforms not only the baseline but also the number of previous MHQA methods such as Longformer, IRC, DFGN, RAG, and HUG and demonstrates comparable answer F1 scores to ETC, BigBird, and ReadTwice while utilizing a smaller number of parameters (see Appendix A). From a practical point of view, compared to the other models with similar performance, our method avoids additional pre-training after warmstarting from a public RoBERTa checkpoint, contributing to enhanced resource efficiency. For example, the ETC paper provides an ablation study showing that removing the pre-training task degrades the model's performance in HotpotQA. We believe that this underscores the efficacy of GEMFormer as a competitive solution for MHQA tasks.
>
> Once again, thank you for your thoughtful evaluation of our work.

---

### Meta-Review · Area_Chair_J37V · 2023-09-06

**Recommendation:** 4
**Best Paper Recommendation:** No

**Metareview:**

This is an interesting short paper, proposing a memory mechanism to enhance the multihop question answering (MHQA) capabilities of transformer models.

While it seems like reviewers agree the idea is interesting in general, review scores range from 2 to 4. I have read the reviews and rebuttals carefully, and have given the paper an in-depth read to try and give it a fair assessment.

## Summary
This work introduces GEMFormer (Global Explicit Memory Transformer) to address the task of MHQA. One significant challenge of MHQA is the fact that information relevant for answering the question is scattered across long, disparate documents, which might not fit into the underlying transformer's context window; even when using models with large window sizes, the attention mechanism might lead to interference of local and global information in the token representations of relevant pieces of information, potentially harming model performance.

The proposed MHQA mechanism is very simple:
- memory construction:
  - Long background documents are split into segments fitting into the transformer's context window.
  - the question is combined with each segment, and put through the transformer and LM head in a forward pass.
  - segment tokens with either highest (top-k) or lowest (below a threshold) LM entropy are selected and copied into the global memory.
- MHQA training
  - each segment is combined with the question and global memory, and the MHQA task is trained in standard fashion.

The authors find that of the two proposed entropy-based memory selection methods, selecting low-entropy tokens works best, with the threshold tuned as a hyperparameter on the MHQA dataset.

The method is trained and tested on 3 MHQA datasets, HotpotQA, 2Wiki, and MuSiQue, with a RoBERTa-base model, and is able to outperform the memory-free baseline on each. While the main body does not contain other -- more relevant -- baselines, these are evaluated against in the appendix in order to make room for some ablation studies that provide some interesting insights into GEMFormer behaviour.

The models chosen for comparison in the appendix seem relevant and sensible, using transformer networks of roughly comparable sizes to GEMFormer. GEMFormer compares favourably overall, even though it is not able to outperform some of the more involved baselines.

## Strengths:
- the presented memory construction method is novel, simple, yet seems effective
- the methodology is clearly presented and easy to follow
- GEMFormer is evaluated on a variety of different datasets.
- the conducted experiments are extensive, and seem to support the author's claims that (i) memory can help MHQA performance; and (ii) selecting low-entropy tokens is required to fill the memory with relevant information.

## Weaknesses:
- the entropy threshold for the (better performaing) low-entropy memory selection is a hyperparameter requiring per-dataset tuning.
- results are consistently higher than the baseline, but not to a high degree
- some parts, especially the discussion, can be hard to follow, as new concepts and ablations are introduced unexpectedly.

## Conclusion
In my opinion, the authors' rebuttals to reviews were satisfactory, and cleared up relevant questions with convincing answers.\
I find myself agreeing with the authors that a per-dataset hyperparameter such as the proposed entropy threshold can be regarded as part of dataset-specific finetuning, which is very much standard -- I am not aware of an MHQA method that is able to be trained on one, and then performs its best on all other MHQA datasets.\
Even though no new state of the art is established (SOTA is not an acceptance criterion in itself), I believe the proposed method shows enough novelty with its simple yet effective approach to memory construction and utilisation for the MHQA task.\
Overall, the work seems sound and merits acceptance; excitement is dampened somewhat by the relatively small improvements achieved, but this is a subjective measure separate from soundness.

While the writing could be slightly improved, such as better splitting of the discussion and ablation parts, as well as clarifying why low entropy works (low entropy = high certainty of tokens, conditioned on the question; changing "least uncertain" to "most certain" to avoid a potentially confusing double negative), this should be no problem for a camera-ready version, and the paper is otherwise very easy to follow.

---

### Decision · Program_Chairs · 2023-10-07

**Decision:**

Accept-Main

**Comment:**

This is an interesting short paper, proposing a memory mechanism to enhance the multihop question answering (MHQA) capabilities of transformer models.

While it seems like reviewers agree the idea is interesting in general, review scores range from 2 to 4. I have read the reviews and rebuttals carefully, and have given the paper an in-depth read to try and give it a fair assessment.

## Summary
This work introduces GEMFormer (Global Explicit Memory Transformer) to address the task of MHQA. One significant challenge of MHQA is the fact that information relevant for answering the question is scattered across long, disparate documents, which might not fit into the underlying transformer's context window; even when using models with large window sizes, the attention mechanism might lead to interference of local and global information in the token representations of relevant pieces of information, potentially harming model performance.

The proposed MHQA mechanism is very simple:
- memory construction:
  - Long background documents are split into segments fitting into the transformer's context window.
  - the question is combined with each segment, and put through the transformer and LM head in a forward pass.
  - segment tokens with either highest (top-k) or lowest (below a threshold) LM entropy are selected and copied into the global memory.
- MHQA training
  - each segment is combined with the question and global memory, and the MHQA task is trained in standard fashion.

The authors find that of the two proposed entropy-based memory selection methods, selecting low-entropy tokens works best, with the threshold tuned as a hyperparameter on the MHQA dataset.

The method is trained and tested on 3 MHQA datasets, HotpotQA, 2Wiki, and MuSiQue, with a RoBERTa-base model, and is able to outperform the memory-free baseline on each. While the main body does not contain other -- more relevant -- baselines, these are evaluated against in the appendix in order to make room for some ablation studies that provide some interesting insights into GEMFormer behaviour.

The models chosen for comparison in the appendix seem relevant and sensible, using transformer networks of roughly comparable sizes to GEMFormer. GEMFormer compares favourably overall, even though it is not able to outperform some of the more involved baselines.

## Strengths:
- the presented memory construction method is novel, simple, yet seems effective
- the methodology is clearly presented and easy to follow
- GEMFormer is evaluated on a variety of different datasets.
- the conducted experiments are extensive, and seem to support the author's claims that (i) memory can help MHQA performance; and (ii) selecting low-entropy tokens is required to fill the memory with relevant information.

## Weaknesses:
- the entropy threshold for the (better performaing) low-entropy memory selection is a hyperparameter requiring per-dataset tuning.
- results are consistently higher than the baseline, but not to a high degree
- some parts, especially the discussion, can be hard to follow, as new concepts and ablations are introduced unexpectedly.

## Conclusion
In my opinion, the authors' rebuttals to reviews were satisfactory, and cleared up relevant questions with convincing answers.\
I find myself agreeing with the authors that a per-dataset hyperparameter such as the proposed entropy threshold can be regarded as part of dataset-specific finetuning, which is very much standard -- I am not aware of an MHQA method that is able to be trained on one, and then performs its best on all other MHQA datasets.\
Even though no new state of the art is established (SOTA is not an acceptance criterion in itself), I believe the proposed method shows enough novelty with its simple yet effective approach to memory construction and utilisation for the MHQA task.\
Overall, the work seems sound and merits acceptance; excitement is dampened somewhat by the relatively small improvements achieved, but this is a subjective measure separate from soundness.

While the writing could be slightly improved, such as better splitting of the discussion and ablation parts, as well as clarifying why low entropy works (low entropy = high certainty of tokens, conditioned on the question; changing "least uncertain" to "most certain" to avoid a potentially confusing double negative), this should be no problem for a camera-ready version, and the paper is otherwise very easy to follow.